# Gold Nanoparticle-Based Colorimetric Biosensing for Foodborne Pathogen Detection

**DOI:** 10.3390/foods13010095

**Published:** 2023-12-27

**Authors:** Sang-Hyun Park, Youngsang You

**Affiliations:** 1Department of Food Science and Technology, Kongju National University, Yesan 32439, Chungnam, Republic of Korea; 2Department of Food Engineering, Dankook University, Cheonan 31116, Chungnam, Republic of Korea

**Keywords:** visible-signaling biosensor, localized surface plasmon resonance (LSPR), food quality, food safety, nanotechnology

## Abstract

Ensuring safe high-quality food is an ongoing priority, yet consumers face heightened risk from foodborne pathogens due to extended supply chains and climate change in the food industry. Nanomaterial-based assays are popular and have recently been developed to ensure food safety and high quality. This review discusses strategies for utilizing gold nanoparticles in colorimetric biosensors. The visible-signal biosensor proves to be a potent sensing technique for directly measuring targets related to foodborne pathogens in the field of food analysis. Among visible-signal biosensors, the localized surface plasmon resonance (LSPR) biosensor has garnered increasing attention and experienced rapid development in recent years. This review succinctly introduces the origin of LSPR theory, providing detailed insights into its fundamental principles. Additionally, this review delves into the application of nanotechnology for the implementation of the LSPR biosensor, exploring methods for utilizing gold nanoparticles and elucidating the factors that influence the generation of visible signals. Several emerging technologies aimed at simple and rapid immunoassays for onsite applications have been introduced in the food industry. In the foreseeable future, field-friendly colorimetric biosensors could be adopted in food monitoring systems. The onsite and real-time detection of possible contaminants and biological substances in food and water is essential to ensure human health and safety.

## 1. Introduction

Nutritional components, including carbohydrates, proteins, and lipids, are ingested to supply a variety of essential nutrients to the human body. Currently, governments place significant emphasis on upholding food quality and safety standards to foster human well-being and promote sustainability. The food industry, for instance, requires a novel, efficient, and user-friendly inspection method as an alternative to the current time-consuming, costly, and cumbersome machinery. Surveillance of foodborne outbreaks plays a crucial role in the food industry due to the expanding scope of food distribution. The security and safety of distributed food hinges on the food producers’ capacity to recognize, detect, and track foodborne pathogens [1]. Pathogens transmitted through food and water can lead to a spectrum of infections, with outcomes spanning from mild fever to fatal consequences [2,3,4]. Annually in the United States, an estimated 48 million domestically acquired foodborne illnesses occur, leading to 127,839 hospitalizations and 3040 deaths, with a societal cost exceeding USD 17.6 billion [5]. Food regulatory agencies and manufacturers have diligently adopted numerous initiatives to reduce the risks associated with contamination by foodborne pathogens. These include the implementation of food cold chain systems and the adoption of hazard analysis and critical control point programs [6]. Nonetheless, the food industry still grapples with the challenge of curbing the incidence of foodborne disease outbreaks. Given the globalization of food supply chains, these outbreaks can occur simultaneously on a global scale due to the intricate food distribution systems in place. Consequently, there is a constant potential for escalating health and economic risks [7]. With conventional methods, such as colony counting, enzyme-linked immunosorbent assay (ELISA), and PCR, the onsite and rapid detection of foodborne pathogens is difficult to analyze. The prompt and precise identification of these pathogens is pivotal in preventing the dissemination of foodborne diseases.

Traditionally, pathogens are identified through colony counting and PCR, but these methods have inherent limitations due to their time-consuming, labor-intensive, lab-based, and costly nature [8,9]. The drawbacks of traditional detection techniques have spurred the innovation of biosensors designed to achieve swifter and more sensitive pathogen detection.

In the quest for ensuring food safety, traditional measures play a pivotal role in detecting foodborne pathogens. Among these, infrared spectroscopy (IR), nuclear magnetic resonance (NMR), and surface-enhanced Raman spectroscopy (SERS) each bring their unique advantages and challenges to the table. IR is a non-destructive analytical method that provides insights into the molecular composition of samples. Its versatility extends to both qualitative and quantitative analyses, making it widely applicable across various industries. However, challenges arise due to limited sensitivity in certain situations, requiring caution when detecting trace amounts. Additionally, the method necessitates relatively pure samples to ensure accuracy, posing potential difficulties when dealing with complex matrices containing impurities [10].

NMR is a powerful analytical tool, providing detailed structural information about compounds and offering a comprehensive understanding of molecular arrangements. Its non-destructive nature makes it well-suited for examining delicate food samples without compromising their integrity. NMR exhibits versatility, being applicable to a diverse range of sample types, from liquids to semi-solids, enhancing its utility in various food matrices. Additionally, it allows for quantitative analysis, enabling the determination of concentrations of specific components in a given sample. However, NMR comes with challenges, including limited sensitivity, especially when detecting low concentrations of analytes, which is crucial for trace amounts of foodborne pathogens. The costs associated with NMR instrumentation can be prohibitive, potentially limiting accessibility for smaller laboratories or facilities with budget constraints. Moreover, effective utilization of NMR necessitates specialized expertise in both operating the equipment and interpreting the obtained data, presenting a challenge in less specialized environments [11,12,13,14].

SERS emerges as a standout technique for its unparalleled sensitivity and specificity. Its ability to detect pathogens at low concentrations makes it a valuable asset in the arsenal of food safety analysts. Furthermore, the versatility of SERS becomes evident in its proficiency in analyzing diverse food samples, including those with complex matrices. This versatility positions SERS as a promising tool for addressing the challenges posed by the intricate nature of various food products. However, researchers and analysts need to navigate through intricacies related to potential interference from sample matrices and the meticulous preparation of substrates [15,16,17].

Biosensors, as a cutting-edge analytical approach, are extensively employed for the analysis of food constituents. They typically consist of two primary elements: (1) a bioligand system responsible for recognizing and capturing the target and (2) a signal transducer that converts biological information into a measurable signal [18,19]. These sensor varieties have experienced rapid development as substitutes for traditional analytical techniques, providing swift, highly selective, user-friendly, and cost-effective detection options [20]. Out of the many detection techniques available, biosensors, born from the fusion of molecular biology and material technology, hold great promise due to their remarkable reactivity, sensitivity, and selectivity [21,22]. Hence, immunosensors based on nanomaterials are gaining prominence in the realm of point-of-care testing thanks to their exceptional attributes. Numerous biosensors have demonstrated remarkable sensitivity, including the capability to detect single cells [23]. The main challenges in foodborne pathogen detection revolve around developing cost-effective and straightforward identification techniques that can detect multiple pathogens, exhibit specificity in distinguishing between various bacteria species, and demonstrate sensitivity to detect bacteria in food samples without the need for pre-enrichment [24,25,26].

The integration of nanotechnology can enhance immunoassays, as nanoparticles possess distinctive physical, chemical, and optical characteristics that set them apart from their bulk counterparts [27,28,29,30]. Notably, gold nanoparticles (AuNPs) have attracted considerable interest for their application in optical biosensors, primarily due to their optical properties, which are dependent on their size and aggregation [31]. This relationship has been harnessed to create AuNPs with diverse colors, sizes, and shapes. Generally, AuNPs have found utility in the detection of nucleic acids, proteins, and entire pathogenic cells. The functionalization of AuNPs with antibodies, carbohydrates, phages, aptamers, small molecules, proteins, or nucleic acids offers a wide array of applications [32,33,34,35,36]. Moreover, there are research results utilizing gold nanoparticles coated with polyethyleneimine, cysteine, oligonucleotide, and electrochemical peptide on the surface, serving as probes that react with target substances [37].

This review provides an overview of diverse approaches to bacteria pathogen detection in food and drinking water. These findings serve as a foundation for enhancing food security in the face of potential infectious diseases. Section 1 outlines the fundamentals of the immunoassay method, while Section 2 delves into visible light biosensing for food safety, encompassing bacteria pathogens.

## 2. Localized Surface Plasmon Resonance Principles of the AuNPs

Of the many visible-signal biosensors, specific surface plasmon resonance (SPR) biosensors offer sensitive, real-time, rapid, and label-free detection capabilities [38,39]. SPR produces an optical signal when the valence electronics oscillate and resonate in a solid metal generated by incident light [40,41]. The optical properties of metallic nanoparticles (MNPs) vary significantly based on their morphologies and sizes. In other words, the visible characteristics of MNPs are intricately connected to their SPR attributes. As shown in Figure 1a,b, when the surface plasmon is confined within an MNP that is smaller than the wavelength of the incoming light, the free electrons in the MNP collectively oscillate, a phenomenon known as localized surface plasmon (LSP) [42,43]. Subsequently, as they couple with incident photons of comparable frequencies, the free electron cloud initiates a synchronized oscillation in relation to the positively charged ions in the lattice, leading to the accumulation of polarization charges on the MNP’s surface. This buildup of polarization charges, in turn, generates a Coulomb field that functions as a restoring force, propelling the electrons in the opposite direction and giving rise to the resonance effect (Figure 1a) [44,45]. As a result, localized surface plasmon resonance (LSPR) yields two primary outcomes: (1) a substantial enhancement of the electromagnetic field produced by nanoparticles, peaking at the surface but diminishing rapidly over a short distance; and (2) the optical extinction of MNPs at the resonance frequency, allowing for detection using conventional UV-Vis spectroscopy or other far-field scattering techniques (Figure 1b) [45,46]. The treatment of LSPR calculations and explanations can be found in the literature [47,48]. LSPR biosensors have attracted significant attention, offering several advantages, including reduced detection location limitation and high flexibility [49,50].

Specifically, gold nanoparticles (AuNPs) possess distinctive optical characteristics that make them extensively applicable in the detection of food- and waterborne pathogens using LSPR [52]. The color of the AuNP solution can be altered by modifying its morphology, including aspects like size and shape, as illustrated in Figure 1c,d [51]. Figure 1e–g depict spherical AuNPs ranging from 5 to 50 nm, each exhibiting distinct absorbance peaks between 515 and 545 nm, along with their respective distributions [53]. Altering the shape of the AuNPs can also lead to changes in their SPR properties. Gold nanorods (AuNRs) display dual absorbance peaks—one associated with the transverse band and the other linked to the longitudinal band in the infrared range. The longitudinal band, especially when employed in immunoassays, proves to be more responsive [54]. This shift in absorbance is often adequate to induce a visible color change, rendering the technique well-suited for straightforward and onsite detection [55]. As the size and aggregation of AuNPs increase, there is a noticeable red shift in the peak absorbance, imparting a stable AuNP solution with a red color. In contrast, the aggregated state of AuNPs imparts a purple color (Figure 2a). This color alteration is readily perceivable to the unaided eye.

## 3. Visible-Signal Strategies for Ensuring Food and Agriculture Product Safety

### 3.1. Non-Functionalized AuNPs for Pathogen Detection

Citrate is commonly used to stabilize AuNPs, with citrate-capped AuNPs carrying a negative charge, while cetyltrimethylammonium bromide (CTAB)-capped AuNPs, especially AuNRs, are positively charged. This technique allows for detection without the need for specific functionalization. Many studies have concentrated on the color transition of AuNPs from red to purple, primarily due to their electrostatic aggregation. Wang et al. [56] reported that *Vibrio parahaemolyticus*, which is usually found in contaminated seafood, could be detected based on thiolated phage nanobody-induced aggregation of AuNPs. In the presence of *V. parahaemolyticus* in the sample, the thiol groups in the phage did not trigger the aggregation of AuNPs. Nevertheless, the thiol group could induce AuNP aggregation when the thiolated phage nanobody was part of a sample lacking the target *Vibrio*. This method could achieve visual detection within 100 min as low as 10^3^ CFU⋅mL^−1^ (Figure 3a,b) [56]. Bu et al. (2019) reported that *Salmonella enterica* serovar Enteritidis and *Escherichia coli* O157 were detected by changing the surface charge of AuNPs. The negatively charged AuNPs were converted into positively charged surfaces using cysteamine and CTAB. The pathogens *S. enteritidis* and *E. coli* O157 were captured by positively charged AuNPs, and the complex interacted with each bacterial antibody using a lateral flow strip (Figure 3c–e) [57]. Therefore, the aggregation of AuNPs occurred at the test line. Pathogens could be detected from 10^3^ to 10^8^ CFU⋅mL^−1^ by the naked eye. Guo et al. (2021) reported a highly sensitive detection method using a bacteria-imprinted polymer and fluorescent and label-free AuNPs within 135 min. Using fluorescence resonance energy transfer, this system achieved highly sensitive detection. In addition, the working range of this method was wide from 10 to 10^7^ CFU⋅mL^−1^ of *Staphylococcus aureus* under optimum conditions [58].

Without the need for nucleic acid amplification, it is advantageous to reduce analysis time and avoid the use of specialized instruments. The use of non-functionalized AuNPs simplifies biosensing assays that produce colorimetric signals. The most crucial limitation of this method is the variety of interferons in the environment, which can cause the nonspecific aggregation of AuNPs and, therefore, produce false signals. The research using non-functionalized AuNPs for the detection of pathogens is summarized in Table 1.

### 3.2. Protein-Functionalized AuNPs for Pathogen Detection

AuNPs can be readily modified with antibodies to facilitate their selective binding to target antigens present on bacterial surfaces. This technique induces the clustering of AuNPs in the vicinity of the target bacteria’s surface due to interactions between antibodies and antigens. An alternative approach involves using the clustering of AuNPs that have been modified with antibodies as a labeling technique to enhance signals by promoting the growth of gold around the original seed particles. As a result, AuNPs have been employed in immune complex (IC) systems as a substitute for ELISA.

A limited level of AuNP aggregation yields an inadequate signal strength, requiring improvement in the signal amplification phase to enhance the signal. Consequently, signal enhancement can be accomplished by introducing a higher concentration of target bacteria into the system. The filtration method can be combined with magnetic nanoparticles for use with complicated samples. This method has been previously used to detect *S. aureus* in milk. First, bovine serum albumin-functionalized magnetic nanoparticles were synthesized and coated with anti-*S. aureus* antibodies. This complex system of magnetic nanoparticles was then added to the target bacteria-contaminated samples. Second, after incubation time, the mixture was filtered to the target 0.8 μm cellulose acetate membrane. The bacteria were magnetically isolated prior to the use of the filtration step for separating the unbound magnetic beads prior to the colorimetric detection. Filtration can separate bacteria attached to magnetic nanoparticles from unbound magnetic nanoparticles because small unbound magnetic nanoparticles can pass through the filtration membrane. Finally, the gold growth solution produced a color change on the surface of the filter [64].

Fluorescent gold nanoclusters (AuNCs) and AuNPs were utilized as rapid, simple, and cost-effective detection systems. In the first step, fluorescent AuNCs were drop-cast onto a fiberglass membrane. *E. coli* O157:H7 antibody-conjugated AuNPs were then loaded into microtubes with fluorescent AuNCs. For detection, *E. coli* O157:H7 samples were placed in microtubes. After a 20 min incubation step, visible sensing was evaluated through Förster resonance energy transfer. Using this method, visible sensing could be achieved from 10^3^ to 10^7^ CFU⋅mL^−1^. Moreover, color recognition could also be achieved using the image sensor of a smartphone. The detection range of the smartphone was from 0 to 10^7^ CFU⋅mL^−1^ [68]. The detection method for whole cells of *Francisella tularensis* was reported by Byzova et al. (2022) [69]. *F. tularensis* could be recognized by monoclonal antibodies in both natural and tap water samples. For visible-signal production, AuNPs of different sizes ranging from 26.6 to 41.8 nm were utilized. The visible detection system consisted of an *F. tularensis* monoclonal antibody and AuNPs. The antibody-conjugated AuNPs could recognize 0–10^7^ CFU⋅mL^−1^ of *F. tularensis* and produce signals within 20 min (Figure 4a,b,d) [68]. This system could detect the presence of *F. tularensis* whole cells at concentrations as low as 3 × 10^3^ CFU⋅mL^−1^ using a color change (Figure 4c) [68]. Moreover, *F. tularensis* lipopolysaccharide was also detected using the same system.

Antibody-functionalized AuNPs can identify pathogens by leveraging their distinct optical characteristics, reducing the need for extensive sample preparation and signal generation. In comparison with ELISA, the detection of pathogens in samples is achieved using paper substrates immobilized with functionalized AuNPs, such as in immune complexes (ICs). This approach is less complex compared with the conventional ELISA, making it easily transportable and demanding only minimal training. Protein-functionalized AuNP methods have certain limitations, e.g., the assay still requires certain laboratory instruments for concentrating the sample, which is frequently only available in the laboratory and requires technical expertise for testing. The methods for pathogen detection using protein-functionalized AuNPs are summarized in Table 1.

### 3.3. Small Molecule-Functionalized AuNPs for Pathogen Detection

AuNPs modified with small molecules are capable of identifying food- and waterborne pathogens through electrostatic, covalent, or receptor-mediated interactions. In such instances, these AuNPs functionalized with small molecules have the ability to cluster around the specified pathogen.

Figure 4e and f illustrate the implementation of a multidetection strategy incorporating nanocomposites [76]. The bacterial probe utilized was the Aptramer-Fe_3_O_4_/MnO_2_ nanocomposite. In the initial stage, the nanocomposite was engaged with the target bacteria, followed by the mixing of gold nanorods with tetramethylbenzidine (oxTMB) for the detection of *E. coli* O157:H7, *S. aureus*, *Listeria monocytogenes*, and *V. parahaemolyticus*. The oxidative activity of the Aptramer-Fe_3_O_4_/MnO_2_ nanocomposite diminished, leading to the conversion of oxTMB to AuNRs and inducing a polychromatic alteration. AuNRs acted as peroxidase mimics, facilitating the oxidation of 3,3′,5,5′-TMB by hydrogen peroxide. Upon combining functionalized AuNRs with the target bacteria in the sample, the functionalized gold nanoparticles aggregated on the surface of the pathogen. Observable color changes were discernible by the naked eye within 40 min [26]. In another study, 13 nm AuNPs functionalized with dithiodialiphatic acid-3aminophenylboronic acid were employed for the detection of *S. aureus*. In this approach, the functionalized AuNPs interacted with *S. aureus* and were subsequently isolated through centrifugation. The separated pathogens appeared red because of the characteristics of the 13 nm AuNPs [70].

In an alternative approach, AuNPs functionalized with sialic acid were employed for the detection of both bacteria and viruses. In the case of influenza viruses, hemagglutinin, present on their surface, was recognized as sialic acid in the host cell during the infection process. The detection of target viruses was achieved through hemagglutinin, as it facilitated the aggregation of sialic acid-functionalized AuNPs. In this approach, a mixture of trivalent α2,6-thio-linked sialic acid-functionalized AuNPs and the influenza virus resulted in a noticeable color change [77].

Utilizing small molecule-functionalized AuNPs facilitates swift and highly sensitive detection. Furthermore, small molecules are more cost-effective compared with other substances like proteins, antibodies, and nucleic acids, making the overall cost of this sensor lower than alternative sensing methods. Nevertheless, small molecule-functionalized AuNPs lack specificity for target pathogens, as some pathogens may produce identical enzymes. Consequently, this approach could yield a false positive response when closely related pathogens are present in a sample. The detection methods employing small molecule-functionalized AuNPs are outlined in Table 1.

To summarize, both functionalized and non-functionalized AuNPs have been employed in colorimetric pathogen detection, targeting surface proteins and whole cells. The objective of this immunoassay is to create methods or devices capable of onsite detection, offering a straightforward visual output. While functionalized AuNPs have contributed to the advancement of simple and rapid immunoassays, these biosensors face challenges in sensitivity when target pathogens are present in complex matrices.

## 4. Conclusions and Prospects

Ensuring the quality and safety of food is crucial for both national governments and food producers, as it directly impacts human health. The effective identification of chemical and biochemical targets in food plays a key role in accurately predicting and diagnosing the health status of food products. In the last few decades, visible-signal biosensors, especially those based on LSPR, have found extensive application in the real-time and onsite detection of food analytes. The sensitivity of LSPR biosensors has been further heightened through the utilization of functional metal nanoparticles with distinctive optical properties, facilitated by advances in nanotechnology. Within the realm of MNPs, AuNPs and AgNPs stand out as the most widely accepted for achieving highly sensitive analyte determination due to their stability and ease of preparation. The application potential of the colorimetric biosensor utilizing AuNPs is promising, owing to its distinctive properties and straightforward implementation. Nonetheless, it encounters several challenges that warrant careful consideration and further investigation. A significant benefit of employing colorimetric biosensors with AuNPs is their capacity to offer a visually clear and easily interpretable signal, rendering them well-suited for onsite and rapid detection. The distinct color alterations stemming from the surface plasmon resonance of AuNPs provide a direct method for detecting the presence of target analytes. Consequently, a majority of these products are presently available to ensure the maintenance of quality and safety in food and agricultural products.

Advancements in recent times have endowed optical biosensors with increased advantages for the analysis of food quality and safety. Nonetheless, there are still several challenges that researchers need to tackle.

The primary challenge confronting researchers in food analysis is achieving ultrasensitive detection. While employing an LSPR optical biosensor for directly determining a single atomic or molecular analyte may not be practical, there is a crucial need for advancements in distinguishing analytes of smaller sizes. Therefore, the sensitivity of the colorimetric biosensor, particularly for low-concentration analytes, merits consideration. Enhancing the detection limits and broadening the dynamic range of AuNP-based biosensors would augment their applicability across a wider array of applications. Moreover, ensuring the stability and reproducibility (selectivity) of LSPR biosensors poses significant challenges. This issue revolves around the potential interference stemming from intricate sample matrices, potentially impacting the biosensor’s specificity and accuracy. Overcoming this challenge entails adjusting the sensor’s selectivity, coupled with a comprehensive understanding of how diverse matrix components can influence the colorimetric response. As outlined in this review, numerous factors contribute to the LSPR peak shift. In this context, preserving the optical biosensor before analysis becomes a concern, especially for LSPR biosensors employing an aggregation mechanism. An additional hurdle is associated with the stability of AuNPs, given that their aggregation or instability over time can influence the reproducibility and reliability of the biosensor. Implementing strategies to bolster the stability of AuNPs and extend their shelf life becomes imperative for the practical deployment of colorimetric biosensors.

In conclusion, although the colorimetric biosensor utilizing AuNPs shows substantial promise, researchers must tackle challenges such as interference from complex matrices, stability concerns, and sensitivity limitations. Surmounting these obstacles will play a pivotal role in unlocking the complete potential of AuNP-based colorimetric biosensors in diverse fields, encompassing environmental monitoring, healthcare, and food safety.

## Figures and Tables

**Figure 1 foods-13-00095-f001:**
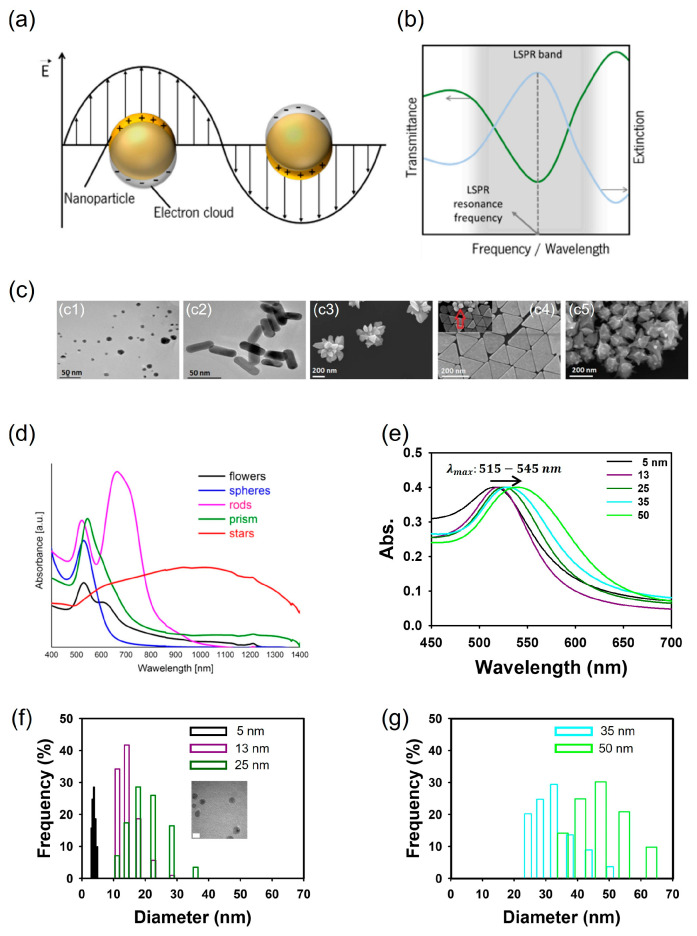
Schematics of localized surface plasmons of metal sphere. (**a**). Schematics of a typical LSPR band, measured in transmittance mode, and extinction spectrum (**b**). Morphology of gold nanosphere (**c1**), nanorods (**c2**), nanoflowers (**c3**), nanoprisms (**c4**), and nanostars (**c5**) (**c**). UV-Vis absorption spectra of gold nanoparticles with different shapes (**d**). Change in absorption spectra of five spherical gold nanoparticles in the range from 5 to 50 nm (**e**) and distribution of gold nanoparticles of diameters 5–25 nm (**f**) and 35 and 50 nm (**g**) synthesized by the citrate reduction of HAuCl_4_. (**a**,**b**) adapted from ref. [45] with permission from MDPI, 2021. (**c**,**d**) adapted from ref. [51] with permission from Springer Nature, 2017.

**Figure 2 foods-13-00095-f002:**
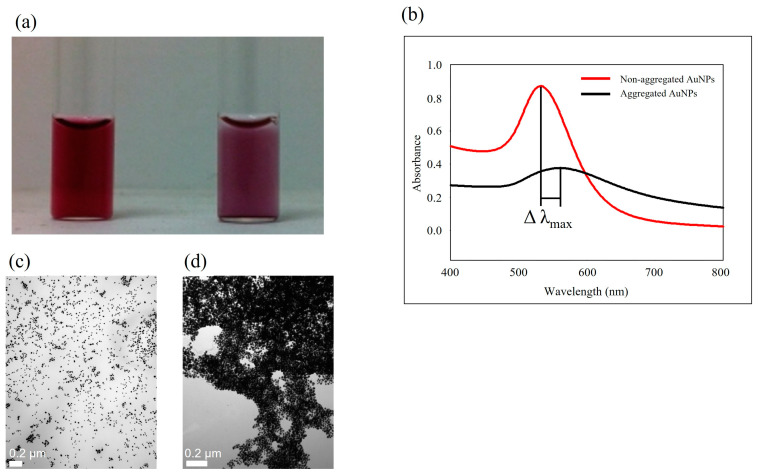
Color of non-aggregated (**left**) and aggregated (**right**) AuNPs (**a**). UV-Vis absorption spectra of aqueous solutions of AuNPs with 13 nm diameter for non-aggregated (red) and aggregated (black) AuNPs (**b**). TEM images of non-aggregated AuNPs (**c**) and aggregated AuNPs (**d**).

**Figure 3 foods-13-00095-f003:**
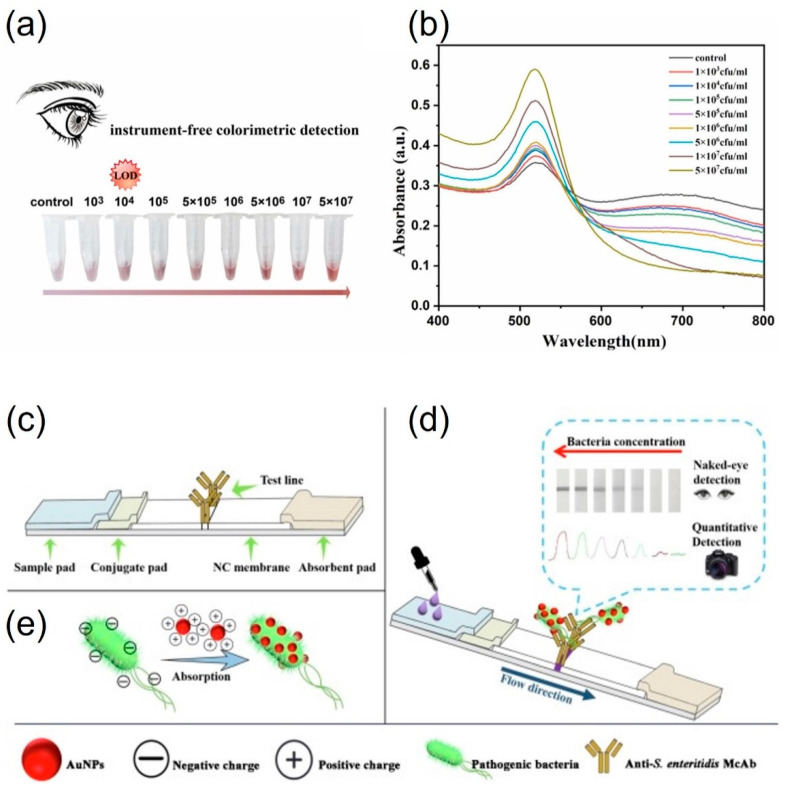
Detection of *V. parahaemolyticus* by the one-step colorimetric immunosensor with (**a**) naked eye and (**b**) UV–vis spectrometry. Schematic illustration of (**c**), the structure of the test strip (**d**), combining AuNPs with *S. enteritidis* (**e**), AuNPs and *S. enteritidis* flow from conjugate pad to absorbent pad, the appearance of the color of the *T*-line, and the principle of quantitative detection strategy of *S. enteritidis* using camera as readout. (**a**,**b**) adapted from ref. [56] with permission from ELSEVIER, 2023. (**c**–**e**) adapted from ref. [57] with permission from ELSEVIER, 2019.

**Figure 4 foods-13-00095-f004:**
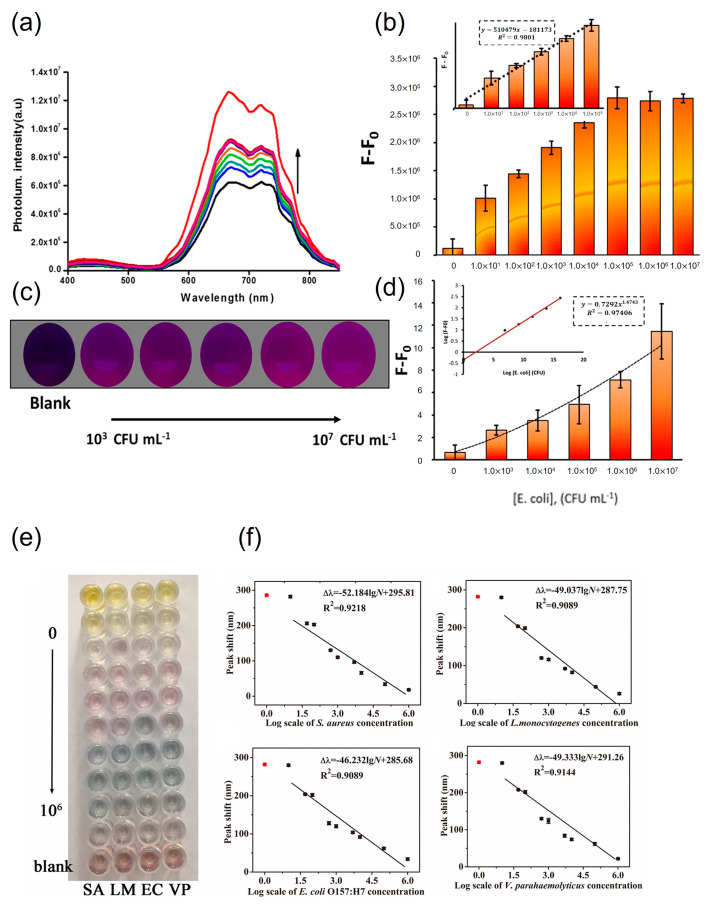
Quantification of *E. coli* O157:H7 using a developed biosensor. (**a**) Fluorescence spectrum of AuNCs/Ab-AuNPs upon addition of different concentrations of *E. coli* from 0 to 10^7^ CFU⋅mL^−1^. (**b**) Calibration curve of the assay using a fluorimeter (F and F_0_ correspond to the fluorescence intensities of the AuNC/AuNP system in the presence and absence of *E. coli*, respectively). (**c**) Typical images of the AuNC/AbAuNP system in the presence of various concentrations of *E. coli* in the range of 0–10^7^ CFU⋅mL^−1^ (0, 10^3^, 10^4^, 10^5^, and 10^6^ CFU⋅mL^−1^ from left to right) captured using a smartphone camera under 365 nm UV light irradiation. (**d**) Calibration curve resulting from processing images in the smartphone. Error bars represent the standard deviation (*n* = 3). Inset: linear behavior of a power function. Photograph (**e**) and calibration curves (**f**), peak shift vs. the logarithm of bacteria concentration of the proposed multicolorimetric assay detecting foodborne pathogenic bacteria at various concentrations (0, 1.0 × 10, 5.0 × 10, 1.0 × 10^2^, 5.0 × 10^2^, 1.0 × 10^3^, 5.0 × 10^3^, 1.0 × 10^4^, 1.0 × 10^5^, and 1.0 × 10^6^ CFU⋅mL^−1^). (**a**–**d**) adapted from ref. [68] with permission from ACS publication, 2020. (**e**,**f**) adapted from ref. [76] with permission from ELSEVIER, 2021.

**Table 1 foods-13-00095-t001:** Summary of alternative food safety detection methods.

Types	Pathogens	Analysis Time	Detection Limit	Working Range	Sample Type	Ref.
Non-functionalized AuNPs	*E. coli* B121and *B. cereus*	~1 h	~10^8^ CFU⋅mL^−1^	ND ^a^	Culture	[59]
	*E. coli* O157:H7 and *S. enterica*	20 min	10^5^ CFU⋅mL^−1^	10^5^–10^8^ CFU⋅mL^−1^	Culture	[60]
	*S. aureus*	~5 min	~1.5 × 10^6^ CFU⋅mL^−1^	ND ^a^	Culture	[61]
	Hepatitis C virus	~30 min	2.5 copies⋅μL^−1^ RNA	~2.5–100 copies⋅μL^−1^	Serum	[62]
	*Enterobacter cloacae* P99	~35 min	16 fmol/mL of P99 β-lactamase	15–80 fmol⋅mL^−1^	β-lactamase	[63]
	*Vibrio parahaemolyticus* (ATCC 17802)	100 min	10^3^ CFU⋅mL^−1^	10^3^–10^7^ CFU⋅mL^−1^	Spiked shrimp	[56]
	*Salmonella enteritidis*	n.d. ^a^	10^3^ CFU⋅mL^−1^	10^3^–10^8^ CFU⋅mL^−1^	Spiked lettuce and pork	[57]
	*S. aureus*	135 min	10^2^ CFU⋅mL^−1^	10^2^–10^4^ CFU⋅mL^−1^	Spiked pork	[58]
Protein-functionalized AuNPs	*S. aureus*	40 min	1.5 × 10^7^ CFU⋅mL^−1^ (milk) 1.5 × 10^5^ CFU⋅mL^−1^ (PBS)	1.5 × 10^7^–1.5 × 10^8^ CFU⋅mL^−1^	Spiked milk	[64]
	*Salmonella enterica* serovar Typhimurium	0.3 h	10^2^ CFU⋅mL^−1^	10^5^–10^8^ CFU⋅mL^−1^	ND ^a^	[65]
	*C. jejuni*	Overnight	10^6^ CFU⋅mL^−1^	10^6^–10^9^ CFU⋅mL^−1^	Culture	[66]
	*G. lambliacysts*	ND ^a^	1 × 10^3^ cells⋅mL^−1^	10^3^–10^4^ cells⋅mL^−1^	Culture	[67]
	*S. enterica serovar*	<5min	10^3^ CFU⋅mL^−1^	10^3^–10^4^ CFU⋅mL^−1^	Culture	[65]
	*E. coli* O157:H7	20 min	10^3^ CFU⋅mL^−1^ 10^0^ cfu⋅mL^−1^ (with smartphone)	10^3^ to 10^7^ cfu⋅mL^−1^	River and tap water	[68]
	*Francisella tularensis*	20 min	10^3^ CFU⋅mL^−1^	10^3^ to 10^9^ cfu⋅mL^−1^	Natural and tap water	[69]
Small molecule-functionalized AuNPs	*S. aureus*	~2 h	50 CFU⋅mL^−1^	5 × 10^2^–5 × 10^6^ CFU⋅mL^−1^	Spiked milk, urine, lung fluid	[70]
	*E. coli* 055:B5 LPS	~5 min	330 fmol⋅mL^−1^ lipopolysaccharides	5–90 pmol⋅mL^−1^	ND ^a^	[71]
	*E. coli* O157:H7	<40 min	7 CFU⋅mL^−1^	7–6 × 10^6^ CFU⋅mL^−1^	Culture	[72,73]
	Influenza B/Victoria and Influenza B/Yamagata	~10 min	0.15 vol% dilution of Hemagglutinati-on assay titer 512 virus	0.15–1.25 vol%	Culture	[74]
	*E. coli* XL1	~10 min	10^2^ CFU⋅mL^−1^ (solution) 10^4^ CFU⋅mL^−1^ (test strip)	10^2^–10^7^ CFU⋅mL^−1^ (solution) 10^4^–10^8^ CFU⋅mL^−1^ (test strip)	Culture	[75]

^a^: not detected.

## Data Availability

Data are contained within the article.

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
