# Peer review of "Gold Nanoparticle-Based Colorimetric Biosensing for Foodborne Pathogen Detection"

_foods, 2023, doi:10.3390/foods13010095_

Round 1
Reviewer 1 Report
Comments and Suggestions for Authors
This manuscript reviewed gold nanoparticle based colorimetric biosensing for food safety, which explored the principles of gold nanoparticle based colorimetric biosensors and the methods of the sensors to detect pathogens in food and drinking water. However, it is too brief and lacks more in-depth discussion. Finally, the manuscript provides some summaries and future perspectives, but unfortunately does not provide a relevant summary of the previous content.
1. Title: the title of this manuscript is “gold nanoparticle based colorimetric biosensing for food safety”. Is this title too simple and too broad in scope? Regarding food safety, the manuscript only mentions some detection methods for pathogens.
2. Introduction: the current status of colorimetric biosensors based on gold nanoparticles is not discussed. Besides, the state-of-art references can be cited.
3. In the title of Section 2, “AuNPs” should be replaced by “AuNPs”, the same below.
4. The formats of the same units in the Table 1 are not unified. In addition, Table 1 can be placed after the content of Section 3, and a simple summary can be added.
5. For the three types of AuNPs detection methods mentioned in the manuscript, there is a lack of more in-depth discussion, including but not limited to the scopes of application and advantages and disadvantages.
6. In the Conclusions, there is a lack of relevant summary of the previous content, and the Prospects are too brief. The colorimetric biosensor based on AuNPs has good application prospects, but it still faces many challenges.
7. Compared to other food safety detection methods, what are the advantages of colorimetric biosensors based on AuNPs? How much time does it take for testing? Is the pre-processing complicated? These can be explored in more detail.
Comments on the Quality of English LanguageEnglish is ok and only minor editing of English language is recommended.
Author Response
We sincerely appreciate your valuable comments and suggestions, which are very helpful for improving the work regarding our manuscript. We have carefully read the comments and revised the manuscript accordingly. Detailed revisions are highlighted in yellow in the manuscript and listed below in a point-by-point manner. We hope that our revisions have improved the paper such that you now deem it worthy of publication in Foods. Please find detailed responses to your comments on attached combined file (cover letter and revised manuscript, PDF).

Reviewer 2 Report
Comments and Suggestions for Authors
This manuscript reviewed gold nanoparticle based colorimetric biosensing for food-borne pathogens detection. However, the manuscript is not well-structured and well-written. Here gives some suggestions for improvement of the manuscript:
1. The title of the article needs to be revised. “Gold nanoparticle based colorimetric biosensing for food-borne pathogens detection?
2. The abstract section needs to be revised, more information should be mentioned.
3. Introduction section, the traditional measures for the detection of foodborne pathogens should be deeply discussed and compared, such as Infra-red (IR), Nuclear magnetic resonance spectroscopy (NMR), SERS and Mass spectrometry are instances of widely used spectroscopic techniques for the detection of pathogens.
4. The author declared that “Section 2 delves into visible light biosensing for food safety, encompassing pathogens, pesticides, and toxins.” (line 78), However, the pesticides and toxins detection was not reviewed in this study.
5. In fact, a review on a similar topic has been published (Food Control 148 (2023) 109510). Thus, authors should describe the similarities and differences between this review and the other published reviews.
6. “Detection based on phages modified Au-NPs” should be added.
7. Please check and correct the grammatical and spelling errors throughout the article.
Comments on the Quality of English LanguageModerate editing of English language required.
Author Response
We sincerely appreciate your valuable comments and suggestions, which are very helpful for improving the work regarding our manuscript. We have carefully read the comments and revised the manuscript accordingly. Detailed revisions are highlighted in yellow in the manuscript and listed below in a point-by-point manner. We hope that our revisions have improved the paper such that you now deem it worthy of publication in Foods. Please find detailed responses to your comments on combined file (cover letter and revised manuscript, PDF).

Reviewer 3 Report
Comments and Suggestions for Authors
Manuscript ID: foods-2758146 by Park et al reviewed the scientific literature on the use of nanoparticles for the detection of food related pathogens. The review was well written; however, some section of the manuscript some additional editing to improve the narrative.
Lines 28-32: On the narrative of foodborne outbreaks, please include in this section the economical burden to the foodborne-related illnesses.
Line 45: Please change the verb tense to “are identified” since these methods are currently used by regulatory agencies.
Line 63: Change to “bacterial species”.
Line 74: Which pathogens? Are these bacterial pathogens or viral pathogens?
Line 78: The current review did not discuss detection of pesticides and toxins. Please delete from the text or add the relevant sections describing detection of pesticides and toxins.
Lines 143-144: Delete “(S. Enteritidis)”, and rewrite as Salmonella enterica serovar Enteritidis. The second time that this pathogen is mentioned in the text, it can then be mentioned as S. Enteritidis. The same principle applies to E. coli O157. In line 146, it would be described as Escherichia coli O157 since it is the first time this species is mentioned in the text.
Line 153: Delete (S. aureus). Capitalize Staphylococcus.
Line 167: Table 1 has the bacterial genus spelled out. Please be consistent and write the genus for the bacterial species since more are included in Table 1.
Lines 178-190: Please revise the narrative to better explain the findings of reference #49 by Sung, Y.J. et al (DOI: 10.1016/j.bios.2012.12.052.). As described by Sung et al, the bacteria were magnetically isolated prior to the use of the filtration step for separating the unbound magnetic beads prior to the colorimetric detection.
Line 199: As already stated for the comment on lines 143-144, delete “(F. tularencis)”.
Author Response

(The authors gave the same response as above.)

Round 2
Reviewer 1 Report
Comments and Suggestions for Authors
Thanks the authors for implementing all the given comments. The manuscript is now in a publishable state.
Author Response
We sincerely appreciate your valuable comments and suggestions, which are very helpful for improving the work regarding our manuscript. We hope that our revisions have improved the paper such that you now deem it worthy of publication in Foods.

Reviewer 2 Report
Comments and Suggestions for Authors
The present manuscript have gained significant improvement after revision. Here gives some suggestions for further revision:
1. The published review (Food Control 148 (2023) 109510) should be cited and discussed in the manuscript.
Author Response
We sincerely appreciate your valuable comments and suggestions, which are very helpful for improving the work regarding our manuscript. We have carefully read the comments and revised the manuscript accordingly. Detailed revisions are highlighted in green in the manuscript We hope that our revisions have improved the paper such that you now deem it worthy of publication in Please find detailed responses to your comments on combined file (cover letter and revised manuscript, PDF).
